# Comparing the effectiveness of environmental DNA and camera traps for surveying American mink (*Neogale vison*) in northeastern Indiana

Eleanor L. Di Girolamo[1], Mark A. Jordan[1], Geriann Albers[2], Scott M. Bergeson[1]*

1 Department of Biological Sciences, Purdue University Fort Wayne, Fort Wayne, Indiana, United States of America, 2 Division of Fish and Wildlife, Indiana Department of Natural Resources, Bloomington, Indiana, United States of America

* bergsos@pfw.edu

**Data Availability Statement:** All relevant data are within the manuscript and its Supporting information files.

## Abstract

While camera traps can effectively detect semi-aquatic mammal species, they are also often temporally and monetarily inefficient and have a difficult time detecting smaller bodied, elusive mammals. Recent studies have shown that extracting DNA from environmental samples can be a non-invasive, alternative method of detecting elusive species. Environmental DNA (eDNA) has not yet been used to survey American mink (*Neogale vison*), a cryptic and understudied North American mustelid. To help determine best survey practices for the species, we compared the effectiveness and efficiency of eDNA and camera traps in surveys for American mink. We used both methods to monitor the shoreline of seven bodies of water in northeastern Indiana from March to May 2021. We extracted DNA from filtered environmental water samples and used quantitative real-time PCR to determine the presence of mink at each site. We used Akaike's Information Criterion to rank probability of detection models with and without survey method as a covariate. We detected mink at four of the seven sites and seven of the 21 total survey weeks using camera traps (probability of detection ($\rho$) = 0.36). We detected mink at five sites and during five survey weeks using eDNA ($\rho$ = 0.25). However, the highest probability of detection was obtained when both methods were combined, and data were pooled ($\rho$ = 0.47). Survey method did not influence model fit, suggesting no difference in detectability between camera traps and eDNA. Environmental DNA was twice as expensive, but only required a little over half (58%) of the time when compared to camera trapping. We recommend ways in which an improved eDNA methodology may be more cost effective for future studies. For this study, a combination of both methods yielded the highest probability for detecting mink presence.

## Introduction

Obtaining species occupancy and distribution data is fundamental to make informed decisions regarding wildlife management. However, gathering reliable data can be difficult for cryptic,

**Funding:** This project was funded in part by a Federal Aid in Wildlife Restoration grant (F23AF02947; W-51-R-03) in cooperation with the Indiana Department of Natural Resources, Division of Fish & Wildlife and the U.S. Fish & Wildlife Service, Wildlife and Sport Fish Restoration Program. The funder had no role in study design, data collection and analysis, nor the decision to publish. However, an agent of the funder (G.A.) contributed to the preparation of the manuscript.

**Competing interests:** The authors have declared that no competing interests exist.

elusive, and rare species and often results in expensive and time-consuming survey efforts [1]. As a result, these species are less frequently surveyed which inherently limits the amount of information that is available to help inform management decisions. The American mink (*Neogale vison*; mink hereafter) is notoriously difficult to survey due to its semi-aquatic nature and elusive behavior [2]. This makes it difficult for researchers and managers to collect enough data to conserve and manage the species in both its native and introduced ranges [3–5].

Mink are semi-aquatic mammals found throughout North America in forested areas near bodies of water with vegetative cover and helophytic vegetation [6]. The species usually occurs at low densities throughout its distribution, in part due to male territoriality [2, 6, 7]. Males tend to exclude other males from their territories and hold larger home-ranges in comparison to females [6, 8]. Despite occurring at low densities, the species has important ecological and economic functions. As a mesopredator, mink can act as a driving force of community structure by the intense predation effort they place on small vertebrate populations, causing a greater impact on primary food sources (e.g., members of Cricetidae, salmonids, and other small fish; [3, 9, 10]) than secondary sources (e.g., crustaceans, birds, amphibians, and insects [11]). Mink is also a desirable pelt in the fur trade. The Association of Fish and Wildlife Agencies reported a national total of 31,814 mink furs harvested in 2018, over half of which were from the Midwest United States (Albers [Unpublished]). Between 2007 and 2017, licensed fur buyers in Indiana reported purchasing an average of more than 1,500 mink pelts per year. However, the number of pelts sold in Indiana has been steadily dropping since 2012 (S1 Fig). This decline potentially suggests that the species may be declining in the state, however, collection of further effort data is required. Mink farms, a common source of fur in the United States, may also be of conservation concerns with regard to wild mink populations, as they can potentially be a source of Aleutian disease and hybridization between wild and escaped farm mink may contribute to genetic homogenization and the introduction of maladaptive gene complexes [12]. American mink pose a conservation threat worldwide, including South America and Asia [13, 14]. Mink are also an invasive species in Europe, where they outcompete the endangered, and virtually identical, European mink (*Mustela lutreola*) causing further population decline [13]. Due to their ecological and economic importance, wildlife managers need effective ways to track changes in mink populations that are independent of harvest, as these numbers can be biased by trapping and trapper behavior [14–16]. Agencies that manage furbearers often have a lack of temporal and monetary resources, which emphasizes the additional need for a time- and cost-efficient method of monitoring mink populations.

Current methods for determining mink occupancy include live-trapping, radio-tracking, scat/track surveys, and camera trap surveys [4, 14, 15]. Live-trapping, radio-tracking, and scat/track surveys are more direct methods that require intensive survey efforts. Alternatively, camera trapping is a non-invasive, low-maintenance method that allows continuous survey efforts over an extended period of time and has been successfully used to survey mink and other Mustelids [17–19]. Camera trapping at North American river otter (*Lontra canadensis*) latrines increased detectability for all carnivorous species detected, including mink [17]. However, camera traps can also be ineffective, especially when surveying for secretive, small, or rare species [20]. For example, low image quality can reduce the ability to correctly identify species, particularly for images collected at night [20, 21] when mink are typically active [22]. Although camera traps are often useful for wildlife management, there may be other more cost- and time-effective methods for surveying species presence.

Environmental DNA (eDNA) is a relatively new method of determining species presence by extracting DNA from environmental samples (e.g., soil and water) [1, 23, 24]. Animals that occupy or have recently visited a sampled area will leave behind DNA in the form of feces, saliva, urine, and skin cells in the environment [25–28]. The persistence of DNA in

environmental samples is largely dependent on the sample type [29]. However, many studies focus on the use of eDNA to monitor fully aquatic species, such as the Eastern Hellbender (*Cryptobranchus alleganiensis*) [30, 31], Asian Carp (*Hypophthalmichthys* spp.) [32], and African Jewelfish (*Hemichromis letrouneuxi*) [25]. Although eDNA is able to persist in aquatic environments for up to 25 days, studies have found that the probability of detection decreases significantly by day 7 [23, 33, 34]. Thus, if eDNA can be detected, surveyors can be confident that the animal occupied the area recently [25]. This method allows areas to be surveyed quickly with minimal disturbance and relatively high certainty. When compared to traditional survey methods (e.g., camera trapping, live-trapping grids, pitfall trapping, and mist netting), one study found that eDNA required less than half of the time in the field and detected 14% more mammal species than camera trapping, which was the second leading method [1]. Specifically, 12 carnivore species were surveyed using a combination of eDNA and camera trapping; including members of the Felidae (*Leopardus pardalis*, *Leopardus wiedii*, *Panthera onca*, *Puma concolor*, *Puma jaguarundi*), Canidae (*Atelocynus microtis*, *Speothos venaticus*), Mustelidae (*Lontra longicaudis*, *Eira barbara*, *Galicitis vittata*), and Procyonidae families (*Potos flavus*, *Procyon cancrivorus*). *L. longicaudis* was the only species detected by the researchers using exclusively eDNA. *P. onca* and *P. cancrivorus* were detected by both methods. The rest were detected exclusively by camera traps [1]. The collection of hair samples by hair tubes is also an effective way of detecting rare carnivore presence [35, 36]. The combination of hair tubes and camera trapping yielded the greatest detection probability (0.85) when compared to the detection probability of any other two-method combination of camera traps, hair tubes, live traps, and eDNA (0.77) [36].

Although the use of eDNA has become more widespread, species specific detection of organisms that are not fully aquatic, like the American mink, is relatively lacking. eDNA has been successfully used to detect the semi-aquatic Eurasian otter (*Lutra lutra*) [37, 38], North American river otters (*Lontra canadensis*) [24], as well as European mink (*Mustela lutreola*) [36]. These studies indicate potential use of this method to detect the American mink. However, field validation of the technique is necessary. Additionally, due to the sensitivity of eDNA methods, it is important to develop proper protocols modified specifically for target species [39]. Limitations of eDNA can include low DNA persistence within an environment depending on factors affecting the rate of DNA degradation (e.g., UV [40], water temperature [40, 41] exposure to sunlight [42]), assay sensitivity and specificity [38, 43]) and available reference databases [1] for properly identifying the target species. The detectability of eDNA can also be largely dependent on the target species and their activity level, body size, and relative abundance [34].

Our objectives in this study were to determine whether eDNA is a viable method of detecting mink presence in aquatic habitat and to evaluate its effectiveness at surveying mink populations compared to the more widely used camera trap method. To do this, we collected water samples from a variety of water bodies (e.g., wetlands, ponds, and a creek) in northeastern Indiana, USA. We analyzed these water samples for presence of mink DNA and compared our results to those from camera traps that we deployed congruently with water sample collection. Environmental DNA has been demonstrated to be an effective, non-invasive, time-efficient method of surveying for fully- and semi-aquatic species. American mink have been observed to spend between 88–95% of their time within 10 m of the nearest water source [6] and dive up to 189 times per day when hunting for food [44]. Studies have found mink presence to be positively correlated with vegetative cover and negatively correlated with open areas and urbanization [5–7]. Due to their activity patterns and preferred habitat use, camera trapping may not be adequate for thorough surveys. We propose that detecting mink using eDNA collected from water sources will be a more time- and cost-effective method when compared to camera trapping.

## Materials and methods

### Study site

We surveyed mink occupancy at seven sites throughout northeastern Indiana, USA in Allen and Kosciusko Counties (centroid of study area = 41˚ 05' 24" N, 85˚ 03' 36" W). Sites consisted of individual bodies of water including four ponds, two wetlands, and one creek. We chose sites with a variety of water bodies and local habitat characteristics to ensure we tested the ability of eDNA surveys in multiple field conditions. Four of the sites were selected based on anecdotal reports of mink presence (Tri-County Fish and Wildlife Area, Fort Wayne Children's Zoo, Eagle Marsh Nature Preserve, and LC Nature Park). The remaining sites (Dorothy and Ray Garman Nature Preserve, Mackel Nature Preserve, and Lakeside Golf and Bowling) had no known reports of mink and were selected based the presence of suitable habitat (i.e., water bodies surrounded by dense vegetation; [6, 7]) and ease of access. We ensured that each site was ≥ 3.4 km apart (mean home-range for male mink; [45–47]) to reduce the likelihood of surveying the same mink in multiple sites. We required sites to be within one hour of travel time from our lab to minimize the amount of time water samples were in transit.

The sites at Fort Wayne Children's Zoo, LC Nature Park, and Lakeside Golf and Bowling were all small man-made ponds, surrounded by manicured/previously manicured grasses. The site at Tri-County Fish and Wildlife Area was an impoundment pond, surrounded by oak-hickory woodlands, as well as flat upland fields. The Eagle Marsh Nature Preserve site was a portion of a larger pond adjacent to a grass/gravel parking lot and ample amounts of coarse woody debris. The pond sampled at Dorothy and Ray Garman Nature Preserve was a natural pond surrounded by agricultural fields. The only creek that was included in our survey was located at Mackel Nature Preserve and is a tributary of the St. Joseph River. The section of the creek we sampled was at the spot where Little Cedar Creek flowed into Cedar Creek and was surrounded by young deciduous forest.

### Sample collection

We deployed camera traps and collected water samples in the late mink breeding season (March to May 2021) to allow for a conservative estimate of species presence due to reduced population densities [2]. We deployed camera traps at 3–4 sites simultaneously for 6 consecutive days (defined as a survey period) and collected water samples in front of each camera on the seventh day. One trap day is defined as one camera deployed for 24-hours. We then moved cameras to new sites and began the survey cycle again; replacing memory cards and batteries when required. We surveyed each site for three non-consecutive weeks, within an 8-week timeframe. We considered trapping effort to be similar between the methods due to the persistence of eDNA in water. Studies have found that eDNA can be detected for 7–14 days before there is a significant decrease in detectability [23, 33, 34]. Thus, an eDNA sample was considered to have included the available DNA from at least the previous 6 days. Because of this, we believe that this survey design should not bias comparisons of effectiveness between the methods based on weekly survey results. Depending on the site composition and perimeter of the body of water, we deployed 3–5 Browning Strike Force HD Pro X (Browning Arms Company, Morgan, UT, USA) passive infrared trail cameras at each site. We attempted to survey as much of the shoreline as possible with the camera traps available. For three sites, a portion of a larger body of water (e.g., large lake or creek) was surveyed to maximize camera coverage at a specific location that appeared ideal for mink (i.e., dense vegetation, coarse woody debris, close to water [7, 48]). One site was partially visible by the public from a road and easily accessible footpath. Therefore, to avoid possible theft, we only placed three cameras that surveyed approximately

50% of the shoreline in concealed areas within this site. We placed cameras approximately 1 m above the ground and 0.5 m from the shoreline (S2 Fig). The cameras were active for 24 h/day, set to three-photo burst per trigger event, had a 0.22 s trigger speed, and a 1-min delay between each trigger event. While camera trap surveys targeting carnivores tend to include bait or another form of attractant [49, 50], we avoided the use of lures to reduce potential DNA contamination and to make it more comparable to the eDNA surveys [27, 33, 51].

To sample eDNA, we collected ten 1 L water samples from each site per survey week. Collection bottles were sterilized with 20% bleach solution before each use. Whenever possible, we collected water from the riverbank by attaching a bottle to a pole to collect the water samples from the shore to reduce the amount of cross-contamination between sites and prevent loosened sediment from entering the water column. At sites where this was not possible (e.g., water too shallow, shore/bank too steep), we wore waders and used gloved hands to collect water samples. The waders and pole were decontaminated with 20% bleach between each site, and we kept the water samples on ice in transit back to the lab. We brought a 1 L Nalgene bottle of ddH$_2$O in the field to each site and treated it as a normal eDNA sample in order to ensure our field practices did not contaminate any water samples. These were used as field negative controls during quantitative polymerase chain reaction (qPCR). The samples were stored in a -80˚C freezer until filtration.

As no animals were handled during this research and tissue samples were provided by other groups, no specific permits were required for this project. However, access to sites was provided by ACRES Land Trust, LC Nature Park, Little Rivers Wetland Project, Fort Wayne Parks and Recreation, Fort Wayne Children's Zoo, and Indiana Department of Natural Resources.

## eDNA analysis

We filtered water samples collected from our sites to consolidate particulate matter and trap any DNA that may be present in the sample. We inverted each 1 L bottle several times before filtering to homogenize the sample. We vacuum filtered half of each 1 L sample through a 250 mL single-use analytical filter funnel with 0.45 *µm* nitrocellulose filter paper. We extracted DNA off the filters using the Qiagen DNeasy® PowerWater® Kit (QIAGEN Sciences, Germantown, MD, USA) following the procedure provided by the manufacturer [23, 52, 53]. All samples were stored in a -80˚C freezer.

We developed a species-specific assay for detecting eDNA of American mink from water samples by adapting protocols from Padgett-Stewart et al. (2016) [24] and Ratsch et al. (2020) [54]. We used the National Center for Biotechnology Information (NCBI) Primer-BLAST (https://www.ncbi.nlm.nih.gov/tools/primer-blast/) to create potential primer pairs for cytochrome b gene on the mitochondrial genome (*cytb*). We conducted in silico testing by aligning the potential primers with *cytb* sequences for mink and closely related species from Genbank (https://www.ncbi.nlm.nih.gov) in Geneious R11 program (https://www.geneious.com). We used Integrated DNA Technologies (IDT) PrimerQuest and OligoAnalyzer Tool (Integrated DNA Technologies, Coralville, IA, USA) to find a complementary probe and to further evaluate the primer set for the guanine and cytosine content (% GC), melting temperature (T$_m$), primer hairpin (ΔG), and primer dimerization (ΔG). Per the suggestions of the IDT OligoAnalyzer Tool [55], our primers were 15–25 base pairs (bp) that created a 70–200 bp amplicon with a 40–60% GC content and a T$_m$ between 57–63˚C with a maximum difference of 2˚C between primer pairs. We also selected for primers with a ΔG (kcal/mol) more positive than -9.0 to limit hairpin formation and dimerization. For the probe, we selected for a T$_m$ that was 6–8˚C greater than the primers and an annealing temperature (T$_a$) less than 5˚C below the

primers. To further refine the prospective assay, we used Geneious to align the primer and probe with various mink whole genome vouchers and *cytb* genes for mink (Accession # MT410953, EF689073.1, KF990329, and KU146454). The primers and probes were ordered from IDT. The selected primers and probe were tested in vitro against DNA of mammals closely related to mink that are native to Indiana to ensure no cross-amplification would occur. DNA was extracted from muscle tissue samples from a North American river otter (*Lontra canadensis*), American badger (*Taxidea taxus*), long-tailed weasel (*Neogale frenata*), short-tailed weasel (*Mustela erminea*), and several mink individuals using DNeasy® Blood and Tissue Kit (QIAGEN Sciences, Germantown, MD, USA) following the kit protocols and using 25 ng of template DNA. The primer set selected had no cross amplification for related species and amplified for all mink individuals tested. We ran multiple qPCR reactions on a temperature gradient, starting with 2°C increments 55–65°C (based on primer $T_m$), then 0.6°C increments between 65.2–68°C with 25 ng of mink DNA. From this, we determined the optimal $T_a$ was 65.6°C and used this $T_a$ for all in vitro tests. We visualized 2.0 $\mu$L of each PCR product on a 2% agarose gel and chose the temperature that produced the band with the greatest saturation on the gel imager, indicating high amplification of the PCR product.

To quantify the DNA extracted from each water sample, we ran qPCR. We used a synthetic double-stranded DNA oligo (gBlock) of the predicted 198bp *cytb* amplicon as a positive control (Integrated DNA Technologies, Coralville, IA, USA). The initial copies per microliter of the gBlock were calculated using the DNA Copy Number and Dilution Calculator on the ThermoFisher website (https://www.thermofisher.com/us/en/home/brands/thermo-scientific/molecular-biology/molecular-biology-learning-center/molecular-biology-resource-library/thermo-scientific-web-tools/dna-copy-number-calculator.html). We made a series of DNA standards with known copy numbers/$\mu L$ ($1.0 \times 10^1$–$1.0 \times 10^9$) from serial dilutions of the gBlock to determine reaction efficiency and test assay performance. The same gBlock was also used to spike a water sample collected in the field (Garman Nature Preserve) in order to use as a field positive control during qPCR. As stated previously, the field negative control consisted of 1 L of ddH$_2$O in a sterile sample bottle that was brought into the field and treated the same as an environmental sample.

We ran each qPCR reaction in 20 $\mu$L aliquots on a 96-well Thermoscienfic AB-2800/W plate with 1X mix of TaqMan® Environmental Master Mix 2.0 (ThermoFisher Scientific, Waltham, MA, USA), 0.3 $\mu$M of each primer, 0.2 $\mu$M of the probe, 1X concentration of 10X TaqMan® Exogenous Internal Positive Control (IPC) and 50X IPC DNA master mix (ThermoFisher Scientific, Waltham, MA, USA). We included the standard curve, field positive and field negative control, and no-template control (pure ddH$_2$O) on each plate. All qPCR reactions were run in triplicate. For a field sample to be considered positive, we required 1 of the 3 technical replicates to amplify, which has been shown to be sufficient to determine species presence [56]. All reactions ran at 95°C denaturation for 3 min and then 50 cycles of 95°C for 15 s, 65.6°C for 30 s, and 72°C for 30 s, and held at a temperature of 12°C. All qPCR reactions were run on the Bio-Rad CFX Connect Real-Time System (Bio-Rad Laboratories, Hercules, CA, USA). FAM (mink) and VIC (IPC) fluorophores were used, and the cycle threshold was kept at the default set by the Bio-Rad CFX Manager 3.1 software (Bio-Rad, Hercules, CA, USA). The limit of detection (LOD) was the lowest number of copies that had at least one positive detection and the limit of quantification (LOQ) was the lowest number of copies that could be detected with at least three positive detections [54, 57].

Amplified qPCR products of positive samples were removed from the initial plate and run through a secondary PCR reaction to increase the concentration of product. The products from the secondary PCR reaction were then run on a 2% agarose gel at 120mV for 45 min to separate products from any potential by-products and extracted using a QIAquick Gel

Extraction Kit (QIAGEN, Hilden, Germany). We cleaned the gel extractions using Zymo Clean & Concentrator -5 Kit (Zymo Research Corp., Irvine, CA, USA) to remove any contaminants and excess primer. Purified products were then sent out to MCLAB (Molecular Cloning Laboratories, San Francisco, CA, USA) to undergo Sanger sequencing to confirm the positives. After sequencing, we aligned the samples to the *cytb* primer pair and probe, the 198 bp amplicon, and species vouchers to make sure each aligned properly using Geneious. We also used NCBI Nucleotide BLAST (NCBI, Bethesda, MD, USA) to further ensure that the sequences matched for mink.

To reduce the probability of cross-contamination, sample filtration, DNA extractions, PCR reagents, and the qPCR machine used were in separate labs. PCR reagents were kept in a separate lab from DNA standards and any extraction material. Lab benches and laminar flow hoods were wiped down with 20% bleach before and after each use.

### Data analyses

We used the program PRESENCE (v. 2.13.35) to compare detection probability for each survey period between camera trapping and environmental DNA survey methods [58]. At least one image clearly containing a mink was required to determine presence at a site during a single survey period using camera trap surveys. Similarly, we required at least one of the three qPCR replicates to amplify from a water sample within a survey period for a site to be positive. We created two models of detection probability: constant detection ($\psi(.), \rho(.)$) and survey technique as a covariate ($\psi(.), \rho(method)$) [59]. Since both surveys were conducted in the same timeframe, we left occupancy ($\psi$) as a constant. We used the Akaike's Information Criterion corrected for small sample sizes ($AIC_c$) to rank the models. If the model with method as a covariate ranked higher than the constant detection model (i.e., constant detection model $\Delta AIC_c \geq 2$; [60]), we inferred that the detection probabilities between the two methods were different. To determine the detection probability of both survey methods combined, we considered a survey period to contain a positive detection if either method met the requirement stated above. We compared the detection probabilities of all models to determine which was highest. The pooled data had to be modeled in a separate run, so we did not rank the pooled data set, as the $\Delta AIC_c$ values were not comparable.

### Cost/Time comparison analysis

We compared the efficiency of eDNA and camera trapping by evaluating the time and funds required to conduct both methods during this study. Time spent conducting camera trap surveys included time spent in the field deploying and checking cameras, including travel, and processing images. Based on ten-timed trial runs, we estimated that we could process 23 ± 1.2 images/min. We then extrapolated this to determine the total time spent to process all camera trap data. Time for environmental DNA surveys included travel to field sites, collecting water samples, and analyzing samples in the lab. We calculated total cost per survey method. For camera trapping, this included the cost of cameras, batteries, memory cards, cable locks, travel (gas at $3.00/gallon, based on average gas prices during the study period), and labor for fieldwork and photo processing (based on a $10.00/hour work wage). For eDNA, the cost included supplies used to collect the samples, supplies for filtration, extraction, and qPCR, the cost of sending positive samples out for sequencing, and labor (including travel, fieldwork, and lab time). Equipment used for sample analyses (e.g., PCR and qPCR machines, laminar hoods) were not included in these calculations.

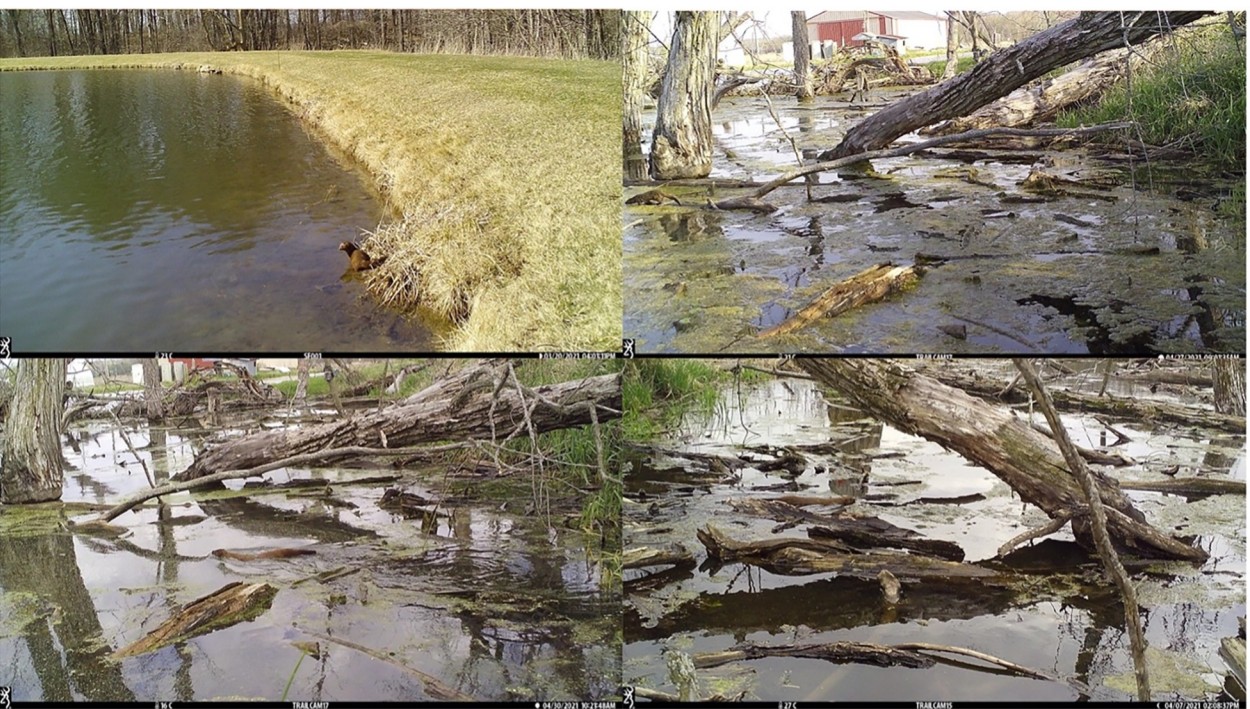

**Fig 1. Camera trap photos showing various American mink (*Neogale vison*) activities detected at sites throughout northeastern Indiana during a survey conducted in March to May 2021 comparing the effectiveness of camera trapping to using environmental water samples for detecting mink presence.** Photos show mink sitting along the shoreline (top left), moving along coarse woody debris (top right), and swimming (bottom row).

## Results

We collected images over a total of 504 trap days. We collected a total of 105,472 usable images (images taken within the 6-day survey period, taken in three-photo bursts). We detected mink at four out of the seven sites (57%), and 12 of the 504 trap days (2%). Mink were primarily detected by cameras located along shorelines adjacent to mature forests. About two-thirds (64%) of the mink detections occurred at night. Out of the images with positive mink detections, observed activities included swimming (50%), moving along course woody debris (43%), and sitting along the shoreline (7%) (Fig 1). We also detected American beaver (*Castor canadensis*), muskrats (*Ondatra zibethicus*), northern raccoon (*Procyon lotor*), coyotes (*Canis latrans*), Canada geese (*Branta canadensis*), white-tailed deer (*Odocoileus virginianus*), and other various bird species (e.g., *Butorides virescens*, *Ardea herodias*, *Agelaius phoeniceus*) (S3 Fig).

We collected a total of 210 water samples (10 samples/site/survey period; 30 total samples/ site) for our analysis of eDNA. Thirteen of these environmental samples (6%) were positive for mink DNA at five of seven sites (71%) and during 5 of 21 survey periods (23.8%) (Table 1). An average of 0.86 ± 0.26 survey periods had positive detections per site and 5 ± 3.06 samples had a positive detection per survey period. Most sites with positive detections had positives from multiple samples taken from different locations within the body of water.

We selected a 21 bp forward (5' TCACTCATATTTGCCGAGACG 3') and a 23 bp reverse (5' CGTAACCTATGAATGCTGTTGCT 3') primer pair to produce a 198 bp amplicon. This primer pair had no cross amplification of closely related species during gel visualization and produced the brightest band for mink (S4 Fig). Within the amplicon, a 29 bp probe (5'- /56-FAM/

**Table 1. Detection (1) and non-detection (0) data for seven sites surveyed for American mink (*Neogale vison*) during the 2021 breeding season between March and May in northeastern Indiana using environmental DNA (eDNA) and camera trapping (CT).**

| Site | Week 1 | | Week 2 | | Week 3 | |
|------|--------|------|--------|------|--------|------|
| | eDNA | CT | eDNA | CT | eDNA | CT |
| CC | 0 | **1** | 0 | 0 | 0 | 0 |
| EM | 0 | **1** | 0 | **1** | **1** | **1** |
| CZ | **1** | 0 | 0 | 0 | 0 | 0 |
| GC | **1** | 0 | 0 | 0 | 0 | 0 |
| GP | 0 | 0 | 0 | 0 | 0 | 0 |
| LC | 0 | **1** | **1** | 0 | 0 | 0 |
| TC | **1** | **1** | 0 | **1** | 0 | 0 |

CC = Mackel Nature Preserve, EM = Eagle Marsh Nature Preserve, CZ = Fort Wayne Children's Zoo, GC = Lakeside Golf Course, GP = Garman Nature Preserve, LC = LC Nature Park, TC = Tri-County Fish and Wildlife Area. Survey periods where the two methods disagreed are shaded in gray and positive detections are in bold.

TCGATATAT/ ZEN/ACACGCAAATGGAGCTTCCA/3IABkFQ/ -3') was selected. We ran each plate with an internal positive control and found no qPCR inhibition for any of the samples. All of these plates resulted in positive detections for all three replicates of the positive controls. There was no amplification in any of the field negative controls (i.e., the ddH$_2$O taken into the field and treated as an eDNA sample), nor the no-template controls (ddH$_2$O), indicating an absence of contamination in the lab. We ran the standard curve in triplicate on each plate ranging from $1.0 \times 10^2$ to $1.0 \times 10^6$ copies, as we had no detections for $1.0 \times 10^1$ copies. We only ran five orders of magnitude to reduce the overall number of reactions and reaction reagents required for each plate. The LoD, where at least one of the three replicates amplified, was $1.0 \times 10^2$ copies, and the LoQ, where all three replicates amplified, was $1.0 \times 10^3$ copies of mink DNA (S5 Fig). The average percent efficiency and R$^2$ for all plates were 97.5 ± 7.88% and 0.94 ± 0.026, respectively. Both of which are within ideal limits stated in the Minimum Information for Publication of Quantitative Real-Time PCR Experiments (MIQE) guidelines [61]. While there were individual plates that did not meet MIQE benchmarks, even the plate with the lowest efficiency (82%) and R$^2$ (0.91) had samples that were positive for mink. Thus, we do not believe that low efficiency or R$^2$ of those plates negatively affected detection for this study. The average C$_q$ value for our field positives was 38.7, which is comparable to the C$_q$ of the 4.0ng/µL of mink DNA extracted from a tissue sample (38.94) (S6 and S7 Figs).

We sent out field positives from four of the five sites with positive detections to an external lab (MCLAB) to be sequenced. We were unable to send the positive sample from one site, as there was an absence of bands in the agarose gel, so the PCR product could not be extracted. Once we received the sequence data back from MCLAB, we aligned the sequences to mink genome vouchers, mink *cytb* sequences, the primer pair, probe, and 198 bp amplicon for the *cytb* gene in Geneious (S8 Fig). Closest off-target species in the BLAST test included the European badger (*Meles meles*) and the marbled polecat (*Vormela peregusna*), with the closest being the marbled polecat at 94% similarity (Table 2). However, neither of these off-target hits were of concern because neither are native to North America [62, 63].

The constant detection model was the top model and the only model within the confidence set in our analysis indicating that survey method did not affect model fit. Therefore, there was no substantial difference between detection probabilities for the two survey techniques (Table 3). The detection probabilities for the camera trap survey and eDNA survey were 0.36 and 0.25, respectively. The naïve occupancy (i.e., proportion of sites with at least one detection;

**Table 2. National Center for Biotechnology information Nucleotide BLAST results for positive environmental DNA samples sequenced in the survey for American mink (*Neogale vison*) in northeastern Indiana 2021.** Listed are the number of base pairs that aligned properly to the forward (F) and reverse (R) primer, whether the F, R, or consensus (C) sequence was used for the BLAST search, the number of base pairs in that sequence, the percent range of hits that came back as mink, number of hits that came back with a percent identification greater than 99% match for mink, and the closest off-target species for each sequence.

| Site | Week | # bp F | # bp R | Sequence Used (F/R/C) | # bp | % ID for *N. vison* | # Hits ≥ 99% ID | Closest Off-Target Species |
|------|------|--------|--------|-----------------------|------|---------------------|-----------------|----------------------------|
| CZ | 1 | 13 | 23 | C | 100 | 98.0–100 | 22 | *V. peregusna* |
| EM | 3 | 21 | 23 | C | 132 | 98.5–100 | 23 | *V. peregusna* |
| EM | 3 | 21 | 23 | C | 133 | 97.7–100 | 23 | *M. meles* |
| EM | 3 | 13 | 23 | C | 124 | 98.4–100 | 23 | *V. peregusna* |
| GC | 1 | 13 | 23 | C | 117 | 98.3–100 | 23 | *V. peregusna* |
| TC | 1 | 21 | 22 | C | 132 | 98.0–100 | 19 | *V. peregusna* |
| TC | 1 | 21 | 22 | C | 132 | 98.5–100 | 24 | *V. peregusna* |
| Field + | 1 | 13 | 23 | F | 103 | 98.1–100 | 19 | *V. peregusna* |

EM = Eagle Marsh Nature Preserve, CZ = Fort Wayne Children's Zoo, GC = Lakeside Golf Course, TC = Tri-County Fish and Wildlife Area.

total detections/number of sites sampled) for mink at our seven survey sites was 0.86. Probability of detection was highest when a combination of the two methods were used (0.48), with mink being detected at six out of seven sites (86%).

The total cost of the eDNA survey was approximately twice that of the camera trap survey (Table 4). The primary cost of eDNA was from consumable lab supplies. The primary cost for the camera trap survey included the price of the camera traps, followed by the labor required for processing all of the camera trap data (S9 Fig). The total time required to conduct the eDNA survey was about 30% less than the time required to conduct the camera trap survey (Table 4). For camera trapping, processing the photos required the most amount of time.

## Discussion

Our objectives for this study were to determine if eDNA is a viable method for surveying American mink and to compare its effectiveness and efficiency to the pre-established camera trap survey method. We found that we were able to successfully detect mink DNA in water sampled from ponds and lakes in Northeastern Indiana. There was overlap between the sites

**Table 3. Akaike's Information Criterion (AIC$_c$), difference from lowest AIC$_c$ ($\Delta$AIC$_c$), AIC weights (AIC$_{wi}$), detection probabilities (P) and standard errors (SE), and number of parameters from the models run in the program PRESENCE for the American mink (*Neogale vison*) survey at seven sites during March-May 2021 in northeastern Indiana.**

| Model | Method | AIC$_c$ | $\Delta$AIC$_c$ | AICw$_i$ | Parameters | P (SE) |
|-------|--------|---------|-----------------|----------|------------|--------|
| $\Psi(.), \rho(.)$ | Constant | 56.25 | 0 | 0.81 | 2 | 0.29 (0.09) |
| $\Psi(.), \rho$ (method) | CT | 60.54 | 4.29 | 0.09 | 3 | 0.36 (0.16) |
| | eDNA | | | | | 0.25 (0.08) |

**Table 4. Estimation of total expenses and time required to survey for American mink (*Neogale vison*) in northeast Indiana using two different survey methods: Camera trapping and environmental DNA.** Both survey methods were conducted within the same period in March to May 2021 during the mink breeding season.

| Survey Method | Total Time (hrs) | Total Cost (USD) |
|---------------|------------------|------------------|
| Environmental DNA | 81.8 | $9,262.99 |
| Camera Trapping | 117.5 | $4,319.96 |

where mink were detected using eDNA surveys and those detected using camera trap surveys. However, eDNA surveys detected mink in two sites that the other method did not while camera traps only detected mink in one novel site. These findings are similar to recent studies which found eDNA increased Eurasian otter detectability at sites by 16–20% when compared to traditional visual methods (i.e., scat/track survey) [38, 64]. Alternatively, we detected mink during more survey weeks using camera traps than using eDNA, which is comparable to a study comparing traditional survey methods (i.e., camera trapping, hair tubes, live traps) and eDNA, which found the probability of detection at each site to be lower when using eDNA (0.66) than camera trapping (0.85) [36]. We determined that the method used did not affect the overall probability of mink detection. An advantage of camera trapping we experienced was being able to collect observations of mink activity throughout the study period, including repeat detections from visiting a site multiple times a day. The collection of this finer temporal scale data would not be possible using eDNA surveys without drastically increasing our sample collection rate and observation of mink activity would not be possible at all. The probability of detection was highest when using a combination of both eDNA and camera trapping, suggesting that surveys employing both methods in tandem may produce the best possible estimate of mink occupancy. However, using a combination of both survey methods would greatly increase the amount of time and funding required, making this study design less feasible for many research groups.

American mink eDNA was detected at five out of the seven sites surveyed but were only detected in 6% of the total environmental samples taken. The low detectability could be due to their large territorial range and semi-aquatic lifestyle, especially in comparison to target species that are fully aquatic and less transient [34]. The amount of DNA present in the environment is affected by the target species (e.g., body mass and activity level) [1, 34] and the amount of time that has passed since the species was present [23, 33, 65]. DNA presence is also dependent on the rate of degradation, which is influenced by various environmental factors, including UV exposure, pH, water temperature, dissolved oxygen, and microbial communities [40, 42, 65]. The water samples collected at one of the sites where mink was not detected (Mackel Nature Preserve) was from a creek. During the study period, there was heavy rainfall that increased the flow rate of the creek and raised the water levels drastically, which could have increased DNA dispersion. A recent study has shown a significant negative relationship between increased stream flow and decreased eDNA copy number [66]. Another study looked at the detection and concentration of Idaho giant salamander DNA following the introduction and removal of salamanders into a previously unoccupied stream. After removing the salamanders from the stream, eDNA concentrations dropped to 0 ng/L and within 24 hours, no eDNA could be detected [42]. False negatives can also occur as a result of the rapid clogging of filter papers which can lead to a concentration of inhibitors [23]. Filtering samples of water with lots of sediment and having filters clog was an issue we also experienced. At Garman Nature Preserve, the water was extremely shallow. To get to an area where the water was deep enough to sample, we had to wade out relatively far, making it difficult to not stir up even more sediment.

While these factors were out of our control, we attempted to prevent false negatives due to low DNA yield by running all samples in triplicate during qPCR [39]. False positives can also be a concern when dealing with eDNA due to the high degree of sensitivity of qPCR. Proper lab practices and sanitation are crucial for preventing any cross-contamination within the lab space [39]. We took many precautions to reduce the possibility of contamination, including conducting PCR and post-PCR phases in separate lab rooms and sterilizing all equipment before and after sample analysis. We also included field negative samples and no-template controls on each PCR plate to certify contamination did not occur. Our LoD and LoQ values are

higher than other eDNA studies report [38, 57, 61, 65], suggesting lower assay sensitivity. However, it is comparable to others with an LoD around 100 copies [54, 67]. This could have been due to human-error in the lab and/or refinements of the lab protocol may be needed to increase assay sensitivity, such as adjusting the primer/probe sequences, ratio of reagents in the qPCR reactions, or annealing temperature. Assay sensitivity is something future studies can build and improve upon. In general, there is a lack of eDNA publications that report LoD/ LoQ, especially with consistent guidelines [68]. When it is included, PCR cycle thresholds tend to be set arbitrarily and the minimum number of replicates at the lowest concentration is highly variable [68]. These variables make it difficult to determine a proper benchmark for LoD/LoQ values in eDNA studies using qPCR.

The four sites that had detections using camera trap surveys had prior anecdotal reports of mink presence. Camera traps did not detect mink at two of the sites where mink presence was confirmed using eDNA. In one of these two sites (Fort Wayne Children's Zoo), we could only deploy enough cameras to cover about half of the body of water as it was within an active zoo enclosure, while wildlife could access both halves of the water body. The number of cameras we were able to deploy in Lakeside Golf Course were also limited by high visibility from the road and easy access of pedestrians to the study site. The lack of thorough camera trap coverage at these sites may have resulted in false negatives for this method. The heavy rainfall that occurred during our study period may have affected our ability to detect eDNA from our creek site at Mackel Nature Preserve by changing the field-of-view for the camera traps (water levels were higher, and in turn, closer to the camera), increasing the water flow. These factors contributed to the camera traps triggering continuously which resulted in the memory card filling up before the end of the survey week. We also had a few instances of image quality affecting mink detection, specifically when the sun was in line with the camera lens and a couple of blurry photos at night.

Based on our survey methods, camera traps cost about half that of eDNA. However, eDNA required almost 30% less time to conduct. Labor time required may have a significant impact on which survey method is best for a study, depending on each unique situation and should not be overlooked. We likely took more water samples than we required for the survey, and filtering redundant water samples greatly increased the overall time we spent on eDNA surveys. Reduced sample collection volume (50 ml per sample) would reduce the total time and cost of eDNA analysis. This would be consistent with several studies showing that eDNA was more time effective when compared to other traditional survey methods (e.g., Sherman live-traps, camera trapping, hair tubes) [1, 36, 69].

While both eDNA and camera trapping are both typically used to collect presence/absence data, they serve unique purposes. Camera trapping can be used to provide information in addition to presence/absence data that investigators may not get from eDNA, such as animal activity patterns, behavior, and community interactions [70]. Alternatively, eDNA can be useful for species that camera trap surveys might have trouble detecting, such as small, cryptic, or elusive animals. eDNA also has the potential of surveying large areas with fewer samples, while camera trap surveys would require intensive set-up to survey a large area thoroughly. In areas where vegetation cover is sparse, such as our site at the abandoned Lakeside golf course, it can be difficult to deploy enough camera traps for a thorough survey safely. For these situations, the use of eDNA may be a better option. We found that using both methods in tandem gave the most comprehensive results in terms of detection and information. Environmental DNA surveys may be most effective at identifying sites containing target species through coarse surveys. This can be followed with more thorough camera trap surveys to collect detailed information.

The use of eDNA to survey for American mink can be particularly useful in areas where feral populations have established such as Europe, Russia, and South America [7]. These feral populations pose conservation concerns to the native European mink, a critically endangered species whose population is decreasing due being outcompeted by American mink [7, 13, 71]. American mink and European mink occupy the same niches, feed on the same food sources, and are cryptic, making visual distinctions extremely difficult [10, 36, 72]. Our study suggests that eDNA may be an effective, non-invasive method of distinguishing occupancy for both mink species. Future studies can focus on the refinement of this assay, specifically to improve assay sensitivity. Once the assay sensitivity is improved, it might be advantageous to test the *cytb* primer and probe pair used in this study against European mink DNA to determine if this assay is species-specific enough to discriminate between the two mink species. Alternatively, designing an assay specific to European mink may be more ideal for management that focuses on the displacement of European mink by American mink in their native range [13]. Monitoring this displacement would help wildlife managers evaluate any conservation plans currently in place for this highly endangered species [13, 71].

## Conclusion

Our study shows that detection probability for American mink was greatest when using a combination of both camera trap survey and eDNA survey. Although environmental DNA was more expensive, it was less time-consuming than camera trapping. Most of the cost for the eDNA study included consumable supplies, meaning it would need to be repurchased for each future study conducted. With refined protocol and laboratory practice, we believe the cost of using eDNA as a survey method can be reduced. These refinements could include determining the minimum number of samples/sample amount required for the detection of the target species, more complete reference sequences in existing databases to reduce the chances of false negatives, and further research into what and how characteristics affect DNA degradation in an environmental setting. Using eDNA to supplement traditional methods, such as camera trapping, can be beneficial to studying any elusive and rare carnivore species. The wide range of environmental samples (e.g., soil, snow, air, water) used to detect DNA allows flexibility when selecting a target species and will provide a more comprehensive survey than using camera trapping alone.

## Supporting information

**S1 Fig. American mink (*Neogale vison*) pelt sales by licensed fur buyers in Indiana between 2007 to 2017.** Dotted line shows linear regression for downward trend. Standard error bars are included for each year (Geriann Albers [Unpublished]).
(DOCX)

**S2 Fig. Photos of how camera traps were set out along the shoreline to survey for American mink (*Neogale vison*) occupancy during the breeding season between March and May 2021 at a pond in the Fort Wayne Children's Zoo.** Site was located in the African exhibit.
(DOCX)

**S3 Fig. Photos showing off-target species detected at a site using camera traps during a survey for American mink (*Neogale vison*) in northeastern Indiana throughout March to May 2021.** Includes photos of a coyote (top left; *C. latrans*), white-tailed deer (top right; *O. virginianus*), a great blue heron (bottom left; *A. herodias*), and a hooded merganser (bottom right; *L. cucullatus*).
(TIFF)

**S4 Fig. Agarose gel of *in vitro* testing of cytochrome *b* gene primer in order to detect American mink (*Neogale vison*) from field samples collected throughout northeastern Indiana.** For each group of six wells, left to right is a negative control, DNA from closely related species, and mink. Each set of six columns/wells were run at a different annealing temperature. The left six wells were run at 66.2˚C, the middle at 65.6˚C, and the right at 65.2˚C. The bands highlighted red indicate areas with saturated signal intensity.
(JPG)

**S5 Fig. Cumulative standard curve for all quantitative PCR plates used in study comparing the effectiveness of environmental DNA to camera trapping in detecting American mink (*Neogale vison*) presence in northeastern Indiana.** There were no detections at or below $1.0 \times 10^1$ copies.
(DOCX)

**S6 Fig. Quantitative PCR data for environmental samples that were positive for American mink (*Neogale vison*) DNA.** Site (CZ = Fort Wayne Children's Zoo, EM = Eagle Marsh Nature Preserve, GC = Lakeside Golf Course, LC = LC Nature Park, TC = Tri-County Fish and Wildlife Area), survey week, and average $C_q$ values are included.
(DOCX)

**S7 Fig. Quantitative PCR run testing a cytochrome *b* primer designed for detecting American mink (*Neogale vison*) and comparing it to closely related species–American badger (*Taxidea taxus*), least weasel (*Mustela nivalis*), long-tailed weasel (*Mustela frenata*), and North American river otter (*Lontra canadensis*).** Mink amplified at $C_q = 38.94$ and there was no off-target species amplification within 50 cycles when using an annealing temperature of 65.6˚C.
(DOCX)

**S8 Fig. Geneious alignment for a positive environmental sample taken during a survey for American mink (*Neogale vison)* in March to May 2021 comparing environmental DNA and camera trapping as survey methods.** From Tri-County Fish and Wildlife Area in northeastern Indiana. The field sample is aligned with *N. vison* genome vouchers, cytochrome *b* (*cytb*) gene, primer set, probe, and amplicon.
(JPG)

**S9 Fig. An itemized breakdown comparing the cost and time required to determine American mink (*Neogale vison*) presence using environmental DNA and camera trapping methods.**
(XLSX)

## Acknowledgments

We thank the Indiana DNR, and the University of Kentucky for tissue samples of various Mustelid species used to test primer specificity. Similarly, we thank the Field Museum of Natural History for providing tissue samples (FMNH #: PL10887, 238703, 235439, 235444, and 178040) for use in this project. We also thank ACRES Land Trust, Fort Wayne Children's Zoo, Fort Wayne Parks and Recreation, LC Nature Park, Little Rivers Wetlands Project, and Tri-County Fish and Wildlife Area for permission to access sites.

## Author Contributions

**Conceptualization:** Eleanor L. Di Girolamo, Geriann Albers, Scott M. Bergeson.

**Data curation:** Eleanor L. Di Girolamo.

**Formal analysis:** Eleanor L. Di Girolamo.

**Funding acquisition:** Scott M. Bergeson.

**Methodology:** Eleanor L. Di Girolamo, Mark A. Jordan, Scott M. Bergeson.

**Project administration:** Scott M. Bergeson.

**Supervision:** Mark A. Jordan, Geriann Albers, Scott M. Bergeson.

**Writing – original draft:** Eleanor L. Di Girolamo.

**Writing – review & editing:** Mark A. Jordan, Geriann Albers, Scott M. Bergeson.

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
