## [Decision Letter · Decision Letter 0]

25 Jul 2024

PONE-D-24-24807Comparing the effectiveness of environmental DNA and camera traps for surveying American mink (Neogale vison) in Northeastern IndianaPLOS ONE

Dear Dr. Bergeson,

Thank you for submitting your manuscript to PLOS ONE. After careful consideration, we feel that it has merit but does not fully meet PLOS ONE’s publication criteria as it currently stands. Therefore, we invite you to submit a revised version of the manuscript that addresses the points raised during the review process. Please consider the comments of Reviewer 1 and include the changes in the manuscript. Thank you for the work done so far, after the inclusion of these minor changes I am happy to accept the manuscript.

We look forward to receiving your revised manuscript.

Kind regards,

Gábor Kemenesi, Ph.D.

Academic Editor

PLOS ONE

Journal Requirements:

3. We note that your Data Availability Statement is currently as follows: All relevant data are within the manuscript and its supporting information files.

Reviewers' comments:

Reviewer's Responses to Questions

**Comments to the Author**

1. Is the manuscript technically sound, and do the data support the conclusions?

Reviewer #1: Partly

Reviewer #2: Yes

2. Has the statistical analysis been performed appropriately and rigorously? 

Reviewer #1: N/A

Reviewer #2: Yes

3. Have the authors made all data underlying the findings in their manuscript fully available?

Reviewer #1: Yes

Reviewer #2: Yes

4. Is the manuscript presented in an intelligible fashion and written in standard English?

Reviewer #1: Yes

Reviewer #2: Yes

5. Review Comments to the Author

Reviewer #1: I have re-reviewed the manuscript by Di Girolamo et al. The authors have adequately addressed most of the concerns raised in the previous review. However, I remain concerned about one point that I raised in the previous review.

I have doubts about the calculation of gBlock copy numbers. In my experience, if the qPCR efficiency is high enough (97.5% in this study), the Ct value for 100 copies per PCR reaction would be around 35. But in this study, the Ct values for 100 copies are much higher than 40, over 45. This is unbelievable in my experience. I suggest the authors to check the copy number of the gBlock again and again.

I also have a question about Fig. S5. It is described that three replicates were used for LOD and LOQ tests. But there are many points in S5 Fig. If the authors have combined the data from different experiments, it is not a good way.

Finally, the amount of DNA (in ng or pg), not the volume, should be described for the specificity test.

Reviewer #2: The authors have made appropriate corrections to the manuscript where it was necessary. I received an appropriate response to my comments.

Excellent work.

Minor comment:

Conclusions, Lines 551-552. Please make this sentence consistent with the results.

6. PLOS authors have the option to publish the peer review history of their article (what does this mean?). If published, this will include your full peer review and any attached files.

Reviewer #1: No

Reviewer #2: No

---

## [Author Response · Author response to Decision Letter 0]

6 Sep 2024

a. Fixed title author affiliations to match sample. Changed body formatting to be single-spaced and fixed level 2 headings to be 16pt font. Fixed figure headings (both figures and supplementary figures) to match correct formatting and fixed file formatting/names.

a. Since no animals were handled during this study, we did not require any permits, although we did obtain permissions from landowners for site access. This statement has been added to the Methods section (Lines 219-222). 

3. We note that your Data Availability Statement is currently as follows: All relevant data are within the manuscript and its supporting information files.

a. We confirm that our submission contains all raw data required to replicate the results of our study.

a. We checked the references list and everything is correct.

From the reviewers:

Reviewer 1:

• I have doubts about the calculation of gBlock copy numbers. In my experience, if the qPCR efficiency is high enough (97.5% in this study), the Ct value for 100 copies per PCR reaction would be around 35. But in this study, the Ct values for 100 copies are much higher than 40, over 45. This is unbelievable in my experience. I suggest the authors to check the copy number of the gBlock again and again.

o On August 30th, 2024, we rechecked our calculations for the gBlock and the serial dilutions. The gBlock ordered was a 250ng desiccate of a 198bp fragment with a molecular weight of 122189.4. It was diluted to 10ng/�L, per the instructions provided by IDT, by adding 25�L of TE pH 8.0, which we calculated to be equivalent to 4.92x1010 DNA copies. From that, we made 200�L of 1.0x109 by adding 4.0�L of 10ng/�L to 196�L of TE. We then made serial dilutions from the 1.0x109 DNA copies down to 1.02x101 DNA copies. We reran our old standard curve, as well as remade the dilutions and ran those dilutions to double check. Our results were consistent as the original manuscript, with the LoD was 1.0x102 and LoQ was 1.0x103, still with Ct values up to almost 46. At 1.0x101, we did not have any detections. 

• I also have a question about Fig. S5. It is described that three replicates were used for LOD and LOQ tests. But there are many points in S5 Fig. If the authors have combined the data from different experiments, it is not a good way.

o Figure 5 was a cumulative standard curve of all standard curves ran on each plate throughout the study. We ran 3-replicates for each dilution for the standard curve ranging from 1.0x102 to 1.0x106. LOD and LOQ were determined from the results of all standard curves ran throughout the study. We can remove this figure, if preferred.

• Finally, the amount of DNA (in ng or pg), not the volume, should be described for the specificity test.

o We changed the volume of DNA to an amount of DNA in line 404.

Reviewer 2:

• Conclusions, Lines 551-552. Please make this sentence consistent with the results.

o Sentences in Lines 551-554 (now lines 574-577) were fixed to be consistent with results.

---

## [Editor Report · Decision Letter 1]

9 Sep 2024

Comparing the effectiveness of environmental DNA and camera traps for surveying American mink (Neogale vison) in Northeastern Indiana

PONE-D-24-24807R1

Dear Dr. Bergeson,

We’re pleased to inform you that your manuscript has been judged scientifically suitable for publication and will be formally accepted for publication once it meets all outstanding technical requirements.

Kind regards,

Gábor Kemenesi, Ph.D.

Academic Editor

PLOS ONE
---

## [Editor Report · Acceptance letter]

12 Sep 2024

PONE-D-24-24807R1 

PLOS ONE

Dear Dr. Bergeson, 

I'm pleased to inform you that your manuscript has been deemed suitable for publication in PLOS ONE. Congratulations! Your manuscript is now being handed over to our production team.

Kind regards, 

on behalf of

Dr. Gábor Kemenesi 

Academic Editor

PLOS ONE